# Deep Learning to Detect Triangular Fibrocartilage Complex Injury in Wrist MRI: Retrospective Study with Internal and External Validation

**DOI:** 10.3390/jpm12071029

**Published:** 2022-06-23

**Authors:** Kun-Yi Lin, Yuan-Ta Li, Juin-Yi Han, Chia-Chun Wu, Chi-Min Chu, Shao-Yu Peng, Tsu-Te Yeh

**Affiliations:** 1Department of Orthopedics, Tri-Service General Hospital, National Defense Medical Center, No. 325, Sec. 2, Chenggong Rd., Neihu District, Taipei 11490, Taiwan; jujilincow@yahoo.com.tw (K.-Y.L.); doc20281@gmail.com (C.-C.W.); 2Department of Surgery, Tri-Service General Hospital Penghu Branch, National Defense Medical Center, Penghu 88056, Taiwan; orthopedicslyt@gmail.com; 3Graduate Institute of Technology, Innovation and Intellectual Property Management, National Cheng Chi University, Taipei 11605, Taiwan; mikehan45@gmail.com; 4School of Public Health, National Defense Medical Center, Taipei 11490, Taiwan; chuchming@web.de; 5Department of Animal Science, National Pingtung University of Science and Technology, Pingtung 91201, Taiwan; sypeng@mail.npust.edu.tw

**Keywords:** deep learning, magnetic resonance imaging, triangular fibrocartilage complex, retrospective study, data analysis

## Abstract

Objective: To use deep learning to predict the probability of triangular fibrocartilage complex (TFCC) injury in patients’ MRI scans. Methods: We retrospectively studied medical records over 11 years and 2 months (1 January 2009–29 February 2019), collecting 332 contrast-enhanced hand MRI scans showing TFCC injury (143 scans) or not (189 scans) from a general hospital. We employed two convolutional neural networks with the MRNet (Algorithm 1) and ResNet50 (Algorithm 2) framework for deep learning. Explainable artificial intelligence was used for heatmap analysis. We tested deep learning using an external dataset containing the MRI scans of 12 patients with TFCC injuries and 38 healthy subjects. Results: In the internal dataset, Algorithm 1 had an AUC of 0.809 (95% confidence interval—CI: 0.670–0.947) for TFCC injury detection as well as an accuracy, sensitivity, and specificity of 75.6% (95% CI: 0.613–0.858), 66.7% (95% CI: 0.438–0.837), and 81.5% (95% CI: 0.633–0.918), respectively, and an F1 score of 0.686. Algorithm 2 had an AUC of 0.871 (95% CI: 0.747–0.995) for TFCC injury detection and an accuracy, sensitivity, and specificity of 90.7% (95% CI: 0.787–0.962), 88.2% (95% CI: 0.664–0.966), and 92.3% (95% CI: 0.763–0.978), respectively, and an F1 score of 0.882. The accuracy, sensitivity, and specificity for radiologist 1 were 88.9, 94.4 and 85.2%, respectively, and for radiologist 2, they were 71.1, 100 and 51.9%, respectively. Conclusions: A modified MRNet framework enables the detection of TFCC injury and guides accurate diagnosis.

## 1. Introduction

Triangular fibrocartilage complex (TFCC) is a vital structure in sports medicine, and its complete observation is difficult because it is situated in a narrow space between the lunate, triquetrum, and ulnar head [1,2,3,4]. According to Ng, A.W., et al., it is difficult to diagnose TFCC injury in MRI with its intrinsic small and thin structure. High-resolution MRI, preferably contrast-enhanced or sometimes with arthrogram, is an ideal detection tool. [5] Observer experience was also an important effect in the diagnosis of TFCC injury in Philip E.’s study, which showed that the difference in the overall accuracy rate for the prediction of a TFCC lesion and its location was 32%. [6] Nowadays, deep learning algorithms for lesion detection allow for the interpretation of medical images with a possibly higher precision and accuracy than the radiologists. [7] Artificial intelligence (AI), especially deep learning algorithms [8,9,10,11,12,13,14,15,16], can identify subtle signs and provide radiologists with more rapid and accurate diagnoses. [11,17,18] For instance, a deep ResNet model, which contains residual links as shortcuts connecting the input and output of a block of convolutional layers [19,20], has been used as the framework to detect the subtle anterior cruciate ligament injuries. [18,21,22] Similarly, an ensemble MRNet has been proposed to detect knee injuries such as cruciate ligaments and meniscal injuries from MRI orthopedic scans. [23] However, there have been no reports using the AI deep learning system to assist in the diagnosis of TFCC injuries on MRI.

As pioneer research, we developed two modified TFCC injury detection algorithms based on the ResNet and MRNet framework and explainable AI to learn MR images from 332 clinical cases and to detect the presence of TFCC injuries.

## 2. Materials and Methods

### 2.1. Study Design

We used the wrist MRI database maintained by the Medical Record Department of the general hospital to conduct a retrospective study involving 332 participants. This study was approved by the Institutional Review Board of our institution (name blinded for peer review). Given its retrospective nature, no patient participated in the design, implementation, and reports of this study, and therefore the requirement for informed consent was waived.

### 2.2. MRI Datasets and Radiologist Reports

From the 332 wrist MRI scans, 143 exhibited TFCC injuries, whereas 189 exhibited healthy hands. The 143 patients with TFCC injuries were diagnosed using arthroscopy following the corresponding MRI examinations. The MRI scans of healthy hands were obtained from patients undergoing routine health checks, and orthopedic surgeons confirmed the absence of TFCC injuries. We retrospectively studied MRI scans obtained over 11 years and 2 months between 1 January 2009 and 29 February 2019. Cases with and without TFCC injuries were randomly assigned to three datasets: 72.8% (*n* = 242) to the training set, 13.6% (*n* = 45) to the validation set, and 13.6% (*n* = 45) to the test set. The size of each dataset was determined based on the minimum requirements for yielding meaningful statistics during and after training. In the internal dataset, we used 13,377 images for the training set, 994 images for the validation set and 3681 images for the test set; in the external dataset, we used 3301 images for the test set (Figure 1).

To compare the differences between the diagnoses provided by AI algorithms and human physicians, we invited two diagnostic radiologists and used the evaluated algorithms to diagnose the MRI scans from the test sets. The task was to determine the presence of TFCC injuries. Among the cases diagnosed with TFCC injuries, we distinguished those where the radiologist detected and classified an injury correctly, detected an injury correctly but classified the injury incorrectly, failed to detect an existing injury, and correctly judged the absence of injury. These results were compared with the AI-generated results for the performance evaluation. The algorithms were evaluated using both the local and external test sets. The external test set was collected at TSGH Song-Shan Branch (Figure 1) and comprised the MRI scans of 12 patients with TFCC injuries and 38 subjects with healthy hands. The scans included coronal T1-weighted, coronal T2-weighted with fat saturation, sagittal proton-density-weighted, sagittal T2-weighted with fat saturation, and axial proton-density-weighted with fat saturation. All scans were acquired in a 3.0 T magnetic field at 3 mm intervals with 16–20 images per slice.

### 2.3. Preprocessing

For image preprocessing, model execution, and performance evaluation, we used Python (version 3.8) programs with the PyDICOM library (version 2.0.0) and OpenCV-Python (version 4.2.0.34) running on a computer equipped with a 2.10 GHz 64 core Intel-Xeon Gold-6130 processor with 276 GB DDR4-SDRAM, 4 T V100 32 GB graphical processing units, and executing a Linux system. The visual data extracted from the DICOM (Digital Imaging and Communications in Medicine) files measured 256 × 256 pixels and were converted into the PNG (portable network graphics) format. To ensure the validity of the analysis, we used a standardization algorithm to convert the pixel values into values between 0 and 255, which is the range of PNG images. 

### 2.4. Algorithms and Model Architectures

We evaluated two framework models, ResNet50 and MRNet, using the PyTorch (version 1.8.0) and TorchVision (version 0.8.1) libraries. The code used in this study is available at https://github.com/akousist/tfcc-detection (accessed on 10 May 2022). Algorithm 1 was based on ResNet50 (Upper part of Figure 2A), which is essential in many applications relying on image classification. We used the ResNet50 algorithm (Figure 2B) to explain the model predictions. ResNet50 contains 16 residual blocks, each consisting of three convolutional layers. Each convolutional layer was followed by batch normalization, and ReLU (rectified linear unit) activation was applied between the convolutional layers and after summing the identity map from the residual link. A convolutional layer was added in front of the ResNet50 architecture to transform eight images along the *z*-axis into one 3-channel 2D image using a design known as 2.5D CNN to incorporate the 3D information. According to the principle of CNN, information can be exchanged between different channels. This is different from the Siamese network-style design of MRNet (described below), where images along the *z*-axis do not exchange information. The effect of integrating *z*-axis imagery can be integrated during model operation. After the z-cropping, the input tensor becomes the size of *n* × 8 × 256 × 256 (*n* is the batch size). This tensor passes through the z-compression network, compresses the z-direction to *n* × 3 × 256 × 256, conforming to the ResNet50 input form, before entering the main architecture of ResNet50. The output of ResNet50 (*n* × 1000) passes through the classifier to obtain a prediction of “with TFCC tears” or “without TFCC tears” (*n* × 2). We applied Adam optimization to accelerate the evolution of stochastic gradient descent. With weights pretrained on the ImageNet dataset (i.e., by applying transfer learning), we used a learning rate of 1 × 10^−5^ and batch size of 16 and trained the model for 20 epochs. A two-class cross-entropy function was employed as the loss function.

Algorithm 2 was derived from MRNet, an ensemble model for diagnosis using a 3D MRI series (lower part of Figure 2A). The MRNet architecture consists of three AlexNets, each of which processes MRI images in one direction (sagittal, coronal, axial). AlexNet was originally a model that processed a single 2D image (Figure 2C). For AlexNet to process the 3D imagery, the batch dimension of the original model was treated as a z-dimension using a Siamese network-like approach, using the same set of model weights to process “a single patient with a set of MRI images in the same scan direction”. Each set of MRI images is first divided into three groups of images according to the scanning direction, then each image will be resized, adjusted to 3 × 256 × 256 (3 means the RGB three color channels), and then each image is arranged in the z-direction into the tensor of s × 3 × 256 × 256 (s is the number of images in the z-direction of the group). The size of this tensor conforms to the input form of AlexNet, which is ready to enter AlexNet. The output of AlexNet becomes a tensor of s × 256. This tensor will enter a z-max pooling to integrate the information of each picture and obtains a 256-dimensional vector. This 256-dimensional vector passes through another classifier to obtain two predictions of “with TFCC tears” or “without TFCC tears”.

### 2.5. Statistical Analysis

Using a pair of validation and test sets, we evaluated the algorithm processing of the MRI scans and the training performance. We evaluated the performance of the final algorithm using the internal test set. To evaluate the model precision, we used the scikit-learn (version 0.19, available at scikit-learn.org) library to calculate the F1 scores and receiver operating characteristic (ROC) curve.

The precision was defined as the number of true positives divided by the number of false positives plus the number of true positives, while the recall was defined as the number of true positives divided by the number of true positives plus the number of false negatives. The F1 score is the harmonic mean of the precision and recall (F1 score = 2/((1/precision) + (1/recall))) and the ROC curve is the softmax function of the absence of a feature.

## 3. Results

### 3.1. Model Performance on the Internal and External Test Sets

The 332 MRI scans were obtained from the same number of subjects (mean age, 36.8 years; 215 men, 64.8%; 117 women, 35.2%). Overall, Algorithm 2 (MRNet framework) outperformed Algorithm 1 (ResNet50 framework) in the detection of TFCC injuries. In Figure 3, Algorithm 1 yielded an AUC of 0.809 (95% CI: 0.670, 0.947) for the TFCC injury detection, as depicted in the receiver operating characteristic curves (ROC curves).

### 3.2. Interpretation and Visualization

An integrated gradient was used after the last convolutional layer of the model, and the results were overlaid on the MRI scans to show the relevance of specific areas for TFCC injury detection. As shown in Figure 4, the algorithm focuses on the region of the contrast media for evaluation and classification. Hence, the Algorithm learned to evaluate the features of the contrast media instead of learning the complete MRI scan with TFCC injury. Figure 4 also shows examples of the integrated grading discrepancies that included MRI scans with poor quality from the original dataset, possibly leading to incorrect predictions.

## 4. Discussion

In our study, two algorithms, MRNet and ResNet50, were proposed to process unlabeled MRI images. Under the classification of only the image data with or without tears of the TFCC, the feature learning of the aforementioned two deep learning algorithms was used to analyze the internal dataset and verify the external dataset. Unlike other studies that spent a lot of time on labeling, this study is the first deep learning exploration on unlabeled MRI images of the TFCC tear.

A well-designed AI-based diagnostic algorithm can reduce human errors in medicine, allowing physicians to provide more precise diagnoses. However, no algorithm has been devised to provide a universal and comprehensive solution for MRI interpretation to date [24,25,26]. The use of deep learning for disease detection in MRI is currently less studied and can be particularly challenging because it often requires the analysis of different image datasets, complicating the process of analysis [17]. In several of Liu, et al.’s studies, labeling data with the CNN training model can improve the accuracy and efficiency of the discrimination between the cartilage and bone injury or cruciate ligament injury in the knee joint [27,28]. To reduce the time costs for labeling and increase the information in the MRI of the TFCC tear, which physicians could miss, we used unlabeled data with deep-learning algorithms to improve these problems.

We developed and evaluated two algorithms based on the ResNet50 and MRNet frameworks to detect TFCC injuries. The results showed that Algorithm 2 with the MRNet framework substantially outperformed Algorithm 1 with the ResNet50 framework. Specifically, Algorithm 2 had a higher AUC of 0.871 than Algorithm 1 (AUC of 0.809). Algorithm 2 was also superior to Algorithm 1 in terms of sensitivity (88.2 vs. 66.7%) and specificity (92.3 vs. 81.5%). These results are attributable to the operation of MRNet, which is based on an ensemble method combining models to solve a single prediction problem. This method of transforming the old model can properly use the existing pretrained weights on the 2D image, and in the case of limited MRI data, the best results can be achieved with the least training resources.

The MRNet architecture in this study was a trained 3D CNN with logistic regression to combine the results for the final prediction. Although training was performed separately and not in an end-to-end approach, the architecture suitably processed MRI scans comprising 3D data from three orientations. In the ResNet50 architecture, we first selected the plane with the most obvious signs for detection, that is, the coronal plane, and obtained eight core images, yielding eight images with resolution of 256 × 256 pixels. Then, we used eight as the feature dimension to treat the scan as a plane with eight RGB channels. By processing the scan through a convolutional layer, we converted it into the common 3 × 256 × 256 dimensions for ResNet-based prediction. We believe that prediction without first combining the algorithm results does not reflect the optimal solution. Therefore, the ResNet50 model can further be improved regarding accuracy.

We used explainable AI to build a glass box model that provides a visual outcome of the algorithm reasoning without resorting to self-interpretable models with less predictive power. Contrast media enhancement in the MRI datasets of TFCC injuries will be focused because of its saliency region. Thus, without labeling by physicians, the algorithm decision on TFCC injury was affected by the contrast media. In other words, the images showed sites of contrast media instead of those of TFCC injuries. Despite our objective being to investigate how explainable AI detects TFCC injury hotspots, hotspots of contrast media were identified. Although using explainable AI to detect TFCC injury hotspots in MR imaging did not yield obvious results, these results may guide future research, and new findings may be unveiled if explainable AI is used to directly detect TFCC injury hotspots in MRI data without the influence of contrast media.

Overall, the proposed algorithms showed high performance in detecting TFCC injuries, which can facilitate diagnosis. However, indicating the TFCC injury sites is challenging if the contrast media provide important contributions to diagnosis. The results indicate that radiologist interpretation is required for AI algorithms to conduct a detailed structural evaluation that identifies soft-tissue injuries in the presence of contrast media. Therefore, the existing deep learning algorithms should be further developed, and the effects of deep learning on diagnostic radiology should be explored in future work.

Some limitations of this study include the following: First, model training was based on the interpretation of radiologists. Although the interpretations of radiologists are the best reference for applications, their interpretations may vary. Second, although the deep learning algorithms considered 3D image evaluation, additional data and patient inquiry are required for a more complete TFCC injury evaluation in clinical practice. Third, the algorithms neglected potential abnormalities associated with hand TFCC such as avascular necrosis and bone abnormalities. Finally, image data obtained from other machine learning algorithms and images from additional external sources were not available, possibly undermining diagnostic performance.

To sum up, we verified the feasibility of applying two deep learning algorithms, the ResNet50 and MRNet frameworks, to detect hand TFCC injuries from features extracted from the MRI scans. The comparison between the performances of the two algorithms further suggests the advantages of the MRNet framework as it works well with 3D images and is competent in integrating the information from the three axes. Overall, we propose an innovative approach on TFCC diagnosis with deep learning algorithms, which we believe will benefit both the radiologists and the surgeons clinically.

## 5. Key Points

An MRNet framework enables the detection of a TFCC injury and guides accurate diagnosis.

The flow direction of the contrast media will affect the convergence of the heatmap.

## Figures and Tables

**Figure 1 jpm-12-01029-f001:**
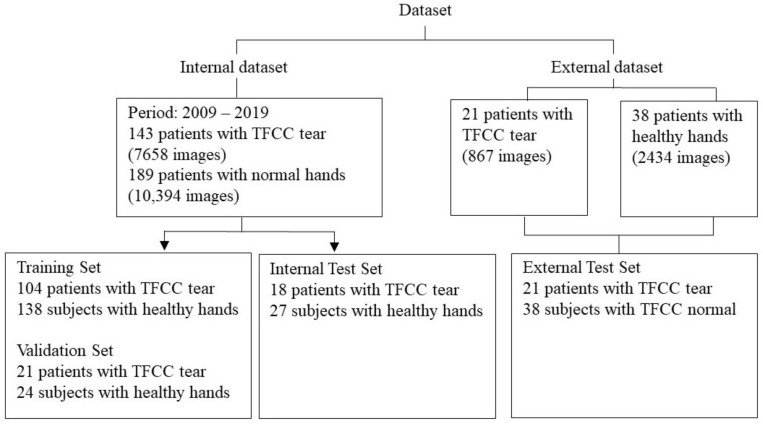
The local and external datasets for model training, validation, and testing.

**Figure 2 jpm-12-01029-f002:**
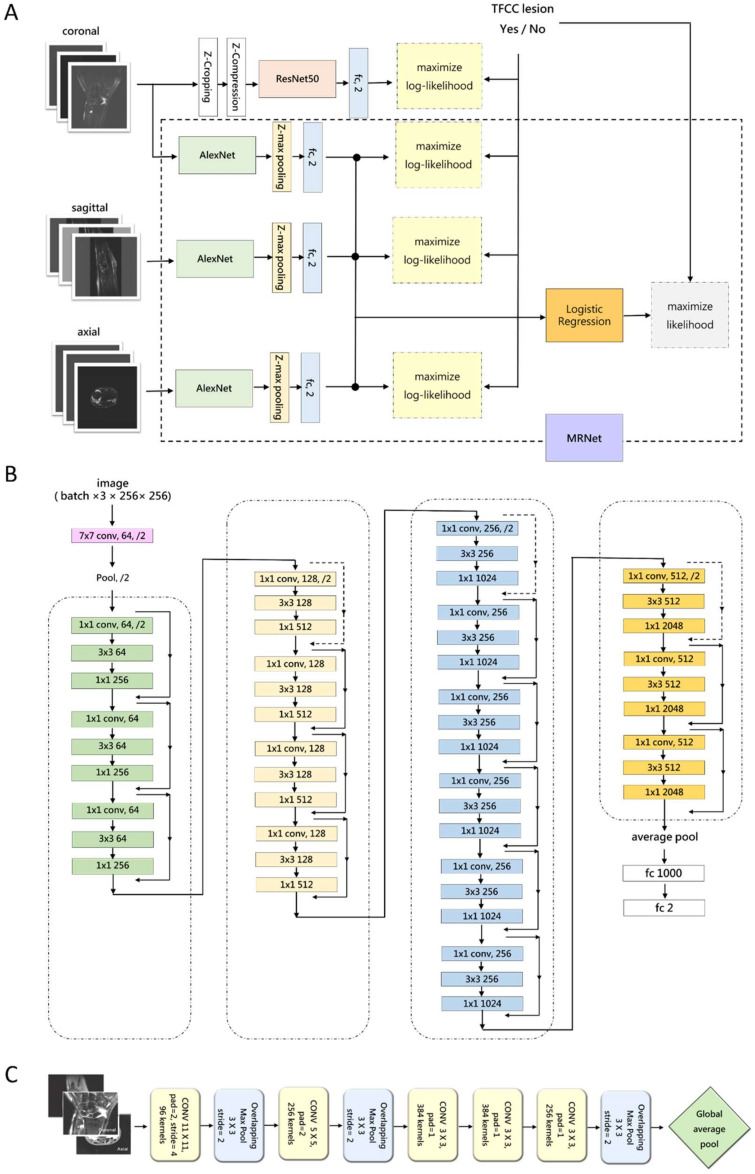
(**A**) An overview of both algorithms used in this study. The topmost data flow shows how Algorithm 1 (ResNet50) processes images along the coronal axis. ResNet50 itself was originally a model for processing 2D images. For ResNet50 to integrate the information from the *z*-axis, the MR images would first go through the z-direction cropping, which selects the middle eight of a set of MR images. Z-compression then transforms the eight images into three channels. In the ResNet50 architecture, this study uses the model’s channel dimension as the z-dimension. Other data flows show how Algorithm 2 (MRNet) processes images along three axes. Training of MRNet is two-stage. Three AlexNets are optimized respectively, indicated by the three log-likelihood maximization. In the second stage training, their outputs are then passed to a logistic regression classifier for an ensemble result. In the design for the MRNet, where the backbone AlexNet was also designed for 2D images, batch dimension was used for the z-dimension, and information along the *z*-axis was integrated with z-max pooling. (**B**)The architecture of ResNet50, the backbone of Algorithm 1 in this study. ResNet50 features residual links, indicated by the jumping arrows to the right of the layer stacks, facilitating the passing of information into a very deep network. Each convolution layer (colored block) is followed by a batch normalization and ReLU activation function, which are not shown in this Figure 2. (**C**) The architecture of AlexNet, the main backbone of MRNet in this study. The feature extraction of AlexNet consists of five convolution layers and three max-pooling layers, in the depicted order. In the original AlexNet, feature extraction ends with global average pooling to 256 × 6 × 6 and the output tensor were flattened to a 9216-dimensional vector for the downstream classifier. Here, we followed the MRNet’s design and average-pool the output to 256 × 1 × 1 before entering the fully-connected network.

**Figure 3 jpm-12-01029-f003:**
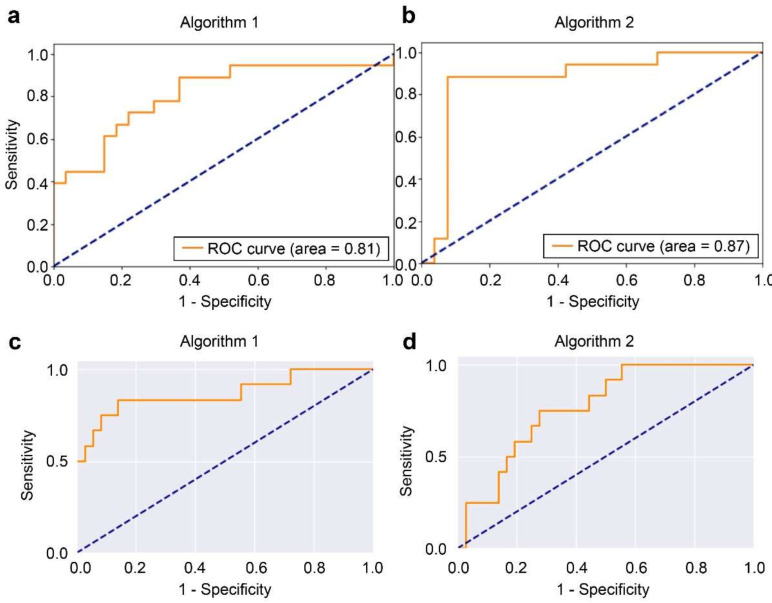
The ROC curves of algorithms (**a**) 1 and (**b**) 2 for the internal dataset. The ROC curves of algorithms (**c**) 1 and (**d**) 2 for the external dataset. ROC, receiver operating characteristic.

**Figure 4 jpm-12-01029-f004:**
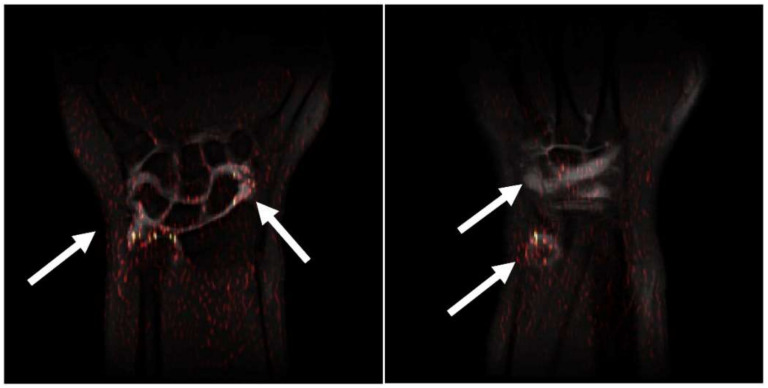
The activation heatmaps overlaid on the original MRI scans. The white arrows indicate the overlapping area of the contrast media and heatmaps.

## Data Availability

Data are contained within the article.

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
