# Peer review of "Deep Learning to Detect Triangular Fibrocartilage Complex Injury in Wrist MRI: Retrospective Study with Internal and External Validation"

_jpm, 2022, doi:10.3390/jpm12071029_

Round 1

Reviewer 1 Report

 I can not find the answers of my previous comments.  Therefore, I have the same comments.

-------------

The major drawback (from my point of view) is that

  • The new contribution/novelty  of this paper in its current form is very marginal and not clear, this is not significant and sufficient to be considered as a full length paper.  

After reading your paper, the following logical questions may arise immediately in a reader’s mind:

  • If the objective is to  detect  triangular fibrocartilage complex injury in wrist MRI, why not use the existing widely used algorithms, there are too many detection algorithms, as it is classically done in different topics?
  • What are the advantages of the proposed algorithms  in this work over classical and existing algorithms?
  • How this work will be beneficial to doctors for effectively and simply detecting triangular fibrocartilage complex injury in wrist MRI?
  • Simply, why there is a need to your study?

In fact, there are too many detection algorithms in different topics.  A good review is needed for the widely used detection algorithms. There is a significant need to a comparison study between the  existing algorithms and the  proposed algorithms in this paper.

The answers of these logical questions help readers to understand the benefits from this study and how doctors will benefit from its findings.

Author Response

Point 1: If the objective is to detect triangular fibrocartilage complex injury in wrist MRI, why not use the existing widely used algorithms, there are too many detection algorithms, as it is classically done in different topics?

Response 1: Actually, previous studies on unlabeled MRI images have not been mentioned or are rarely mentioned. Our study uses two algorithm frameworks, MRNet and ResNet50, which are commonly used in 3D images in deep learning, to perform deep learning of features for unlabeled MRI image data that are only classified, and establish a prediction model.

Point 2: What are the advantages of the proposed algorithms in this work over classical and existing algorithms?

Response 2: These framework of two algorithms, MRNet and ResNet50, were used to process unlabeled MRI images in our study. Under the classification of only the image data with or without tear of TFCC, the feature learning of the aforementioned two deep learning algorithms was used to analyze the internal dataset and validate the external dataset. Without physicians’ labeling,all features are deeply learned by the computer. It’s unlike typical and existing algorithms. Typical algorithm processing requires a large amount of labeling of physicians' knowledge before deep learning is performed. Therefore, the features learned by deep learning are the knowledge of doctors, but there is a lack of a lot of learning about image features. Our study adopts another method to let the computer conduct deep learning autonomously and is the first deep learning exploration with unlabeled MRI images of TFCC tear.

Point 3: How this work will be beneficial to doctors for effectively and simply detecting triangular fibrocartilage complex injury in wrist MRI?

Response 3: Observer experience is also an important effect in the diagnosis of TFCC injury in Philip E.’s study, which showed differences of the overall accuracy rate between observers for prediction of a TFCC lesion and its location were 32%. So, according to our study, we can confirm that high accuracy of the AI-assisted diagnosis of TFCC injury is beneficial to physi.

Point : Why there is a need to your study?

Response 4: To provide an innovative approach on TFCC diagnosis with deep learning algorithms which we believe will benefit both the radiologists and the surgeons clinically.

Reviewer 2 Report

In order to utilize deep learning to predict the probability of triangular fibrocartilage complex (TFCC) injury in patients’ MRI scans, the authors retrospectively studied medical records over 11 years and 2 months, collecting 332 contrast-enhanced hand MRI scans showing TFCC injury or not from a general hospital, in which two convolutional neural networks were used with MRNet (algorithm 1) and ResNet (algorithm 2) framework for deep learning. It is verified that a modified MRNet framework enables the detection of TFCC injury and guides accurate diagnosis. 

Some comments are as follows: 

1) I suggest adding a paragraph highlighting your innovations.

2) The following papers of image processing are recommended to be added in Section 1:

------Hyperspectral Image Segmentation Using a New Spectral Unmixing-Based Binary Partition Tree Representation, IEEE Transactions on Image Processing, 2014

------Fuzzy C-Means Clustering Through SSIM and Patch for Image Segmentation, Applied Soft Computing, 2020.

3) For Figure 2, a note needs to be added.

4) The detailed content needs to be expanded for two convolutional neural networks networks.

Author Response

Response to Reviewer 2 Comments

Point 1: Adding a paragraph highlighting your innovations.

Response 1: In our study, two algorithms, MRNet and ResNet50, are proposed to process unlabeled MRI images. Under the classification of only the image data with or without tear of TFCC, the feature learning of the aforementioned two deep learning algorithms is used to analyze the internal dataset and verifiy the external dataset. Different from other studies that spend a lot of time on labeling, this study is the first deep learning exploration on unlabeled MRI images of TFCC tear.

Point 2: For Figure 2, a note needs to be added.

Response 2: We add a Fig.2b to explain ResNet50 algorithm.

Reviewer 3 Report

Lin et al. trained and compared two Deep Learning models for detecting TFCC injury in wrist MRI. This type of work is common these days but has not been done for TFCC detection from MRI images. Notably, TFCC detection in ultrasound images has recently been reported (PMID: 35447195).

We think the work will be of interest but the authors need to improve several places of presentation. 

1. The current descriptions of the models are minimalist. The authors need to expand those to make the manuscript more self-contained. Although integrated gradients is a well-established method, the authors should properly cite it.

2. We think Fig. 3b has been taken from elsewhere. At least, it shows up as a copyrighted material in a Google Image search.

3. We commend the authors for a candid discussion on how the model picks up contrast media signals as opposed to learning the true TFCC signal. Do they think this is a sample size issue? Could they try image augmentation? 

4. The authors also need to comment on MRI signal variation between sources. How variable are the MRI signals from two different machines? Are the models sensitive to such signals?

Author Response

Point 1: The current descriptions of the models are minimalist. The authors need to expand those to make the manuscript more self-contained. Although integrated gradients is a well-established method, the authors should properly cite it.

Response 1:

We update the citation in the discussion.

Point 2: We think Fig. 3b has been taken from elsewhere. At least, it shows up as a copyrighted material in a Google Image search.

Response 2: We have changed the content of Fig. 3b.

Point 3: We commend the authors for a candid discussion on how the model picks up contrast media signals as opposed to learning the true TFCC signal. Do they think this is a sample size issue? Could they try image augmentation?

Response 3: If the highlight of this paper is placed on the "correct rate", then the contrast media is helpful, because "image with contrast media " can allow the algorithm to learn the doctor's judgment mode. This also corroborates that "in the absence of the contrast media and the absence of a physician's labeling, information of TFCC tear in the MRI could be very rare." It’s not just the issue of sample size. So, we think that without the physician labeling, learning the "MRI with contrast media" is the only way for interpretable AI. In the future, we may modify the algorithm to find features in MRI without contrast-media.

Point : The authors also need to comment on MRI signal variation between sources. How variable are the MRI signals from two different machines? Are the models sensitive to such signals?

Response 4: ResNet forms a bottleneck in z-cropping and z-compression that blocks information consolidation in the z direction. Thus, the feature capture in the z direction is affected, indirectly affecting the accuracy. The amount of image data remains critical for the accuracy of MRNet. The larger the number, of course, the more advantageous the model studied in terms of the diversity of training learning and feature capture. Comparison to Bien et al’s study. (https://doi.org/10.1371/journal.pmed.1002699), the number of images in our project only has about 1/3 of the amount. Moreover, MRNet itself is a parallel model using 2D image, and integrating 3D information with MaxPool at the last layer. It may still have limitations. Therefore, for future study, we will using a full 3D-CNN, which may be the better optimization scheme.

Round 2

Reviewer 1 Report

The revised version looks much better. 

Author Response

Thank you for your incisive opinions.

Reviewer 3 Report

The authors didn’t adequately address our previous point #1. Citing the paper on integrated gradients was one point but more importantly, we recommended the authors to describe the models properly for biologists who need to understand how these models work.

Fig 3: now there are two panels labeled as 3a and 3b is not legible. There is a surprising level of dissimilarity between 3a and 3b: whereas 3b seems to be more detailed in providing the dimensions and the details of the architecture, 3a just shows some tensors. The authors need to do a better job at presenting their models.

The authors didn’t address our last point “The authors also need to comment on MRI signal variation between sources. How variable are the MRI signals from two different machines?” Their response to this last point is a completely irrelevant one.

Author Response

Point 1: The authors didn’t adequately address our previous point #1. Citing the paper on integrated gradients was one point but more importantly, we recommended the authors to describe the models properly for biologists who need to understand how these models work.

Response 1:We extremely appreciate this important comment. The manuscript has been revised to make it more readable for biologists. More details of model’s work are described at below. The changes were highlighted as red color as the new version of manuscript. We have provided new pictures with high resolution as following

We evaluated two framework models, ResNet50 and MRNet, using the PyTorch (version 1.8.0) and TorchVision (version 0.8.1) libraries. The code used in this study is available at https://github.com/akousist/tfcc-detection. Algorithm 1 is based on ResNet50 (Upper part of Figure 2a), which is essential in many applications relying on image classification. We use the ResNet50 algorithm to explain the model predictions. ResNet50 contains 16 residual blocks, each consisting of three convolutional layers. Each convolutional layer is fol-lowed by batch normalization, and ReLU (rectified linear unit) activation is applied between the convolutional layers and after summing the identity map from the residual link. A convolutional layer is added in front of the ResNet50 architecture to transform eight images along the z-axis into one 3-channel 2D image using a design known as 2.5D CNN to incorporate 3D information. According to the principle of CNN, information can be exchanged between different channels. This is different from Siamese network-style design of MRNet (described below), where images along z-axis do not exchange information. The effect of in-tegrating z-axis imagery can be integrated during model operation. After the z-cropping, the input tensor becomes of size n×8×256×256 (n is the batch size). This tensor passes through the z-compression network, compresses the z-direction to n×3×256×256, conforming to the ResNet50 input form, before entering the main archi-tecture of ResNet50. The output of ResNet50 (n×1000) passes through classifier to obtain a prediction of "with TFCC tears" or "without TFCC tears" (n×2). We apply Adam optimization to accelerate the evolution of stochastic gradient descent. With weights pretrained on the ImageNet dataset (i.e., by applying transfer learning), we use a learning rate of 1×10−5 and batch size of 16 and train the model for 20 epochs. A two-class cross-entropy function is employed as the loss function.

Figure 2a. Overview of both algorithms used in this study. The topmost data flow shows how algorithm 1 (ResNet50) processes images along coronal axis. ResNet50 itself was originally a model for processing 2D images. For ResNet50 to integrate information from the z-axis, the MR images would first go through the z-direction cropping, which selects the middle eight of a set of MR images. Z-compression then transforms the eight images into three channels. In the ResNet50 architecture, this study uses the model's channel dimension as z-dimension. Other data flows show how algorithm 2 (MRNet) processes images along three axes. Training of MRNet is two-stage. Three AlexNets are optimized respectively, indicated by three log-likelihood maximization. In the second stage training, their outputs are then passed to a logistic regression classifier for an ensemble result. In MRNet’s design, where the backbone, AlexNet, was also designed for 2D images, batch dimension is used for z-dimension, and information along z-axis is integrated with z-max pooling.

Figure 2b. Architecture of ResNet50, the backbone of algorithm 1 in this study. ResNet50 features residual links, indicated by the jumping arrows to the right of the layer stacks, facilitate passing of in-formation into very deep network. Each convolution layer (colored block) is followed by a batch normalization and ReLU activation function, which are not shown in this figure.

Figure 2c. Architecture of AlexNet, the main backbone of MRNet in this study. Feature extraction of AlexNet consists of 5 convolution layers and 3 max-pooling layers, in the depicted order. In original AlexNet, feature extraction ends with global average pooling to 256×6×6 and the output tensor were flattened to a 9216-dimensional vector for downstream classifier. Here we follow MRNet’s design and average-pool the output to 256×1×1 before entering the fully-connect network.

Algorithm 2 derives from MRNet, an ensemble model for diagnosis using a 3D MRI series (Lower part of Figure 2a). The MRNet architecture consists of three AlexNets, each of which processes MRI images in one direction (Sagittal, Coronal, Axial). AlexNet was originally a model that processed a single 2D image. For AlexNet to process 3D imagery, the batch dimension of the original model is treated as a z-dimension, using a Siamese network-like approach, where the same set of model weights is used to process a set of MRI images from one scanning direction, from one patient. Each set of MRI images will first be divided into three groups according to the scanning direction. Each image will be resized to 3×256×256 (3 means the RGB three color channels). The images of the same scanning direction will be arranged along the z-axis into a tensor of s×3×256×256 (s is the number of images in this group). The size of this tensor conforms to the input form of AlexNet and is ready to enter AlexNet. The output of AlexNet is a tensor of s×256. This tensor will enter a z-max pooling to integrate the information of each scan slice and results in a 256-dimensional vector. This 256-dimensional vector passes through another classifier to get two-way prediction of "with TFCC tears" or "without TFCC tears".

Point 2: Fig 3: now there are two panels labeled as 3a and 3b is not legible. There is a surprising level of dissimilarity between 3a and 3b: whereas 3b seems to be more detailed in providing the dimensions and the details of the architecture, 3a just shows some tensors. The authors need to do a better job at presenting their models.

Response 2:Thank you very much for this recommendation. To avoid the misunderstanding and show the consistency at our models, we erased the Fig.3a and presented more detail information in Fig.2a-2c.

Point 3:The authors didn’t address our last point “The authors also need to comment on MRI signal variation between sources. How variable are the MRI signals from two different machines?” Their response to this last point is a completely irrelevant one.

Response 3:Thank you very much for the comments. The aim of our study is whether it is possible to learn weakly labelled data through an algorithm and build a model to detect whether the TFCC is injured. For external data, we did observe that the algorithm would have different results when detecting data from different hospitals. Currently, we couldn’t explain this issue, but we have already begun to collect data from different regions to find out probable factor in further research.

This manuscript is a resubmission of an earlier submission. The following is a list of the peer review reports and author responses from that submission.

Round 1

Reviewer 1 Report

The major drawback (from my point of view) is that

  • The new contribution/novelty  of this paper in its current form is very marginal and not clear, this is not significant and sufficient to be considered as a full length paper.  

After reading your paper, the following logical questions may arise immediately in a reader’s mind:

  • If the objective is to  detect  triangular fibrocartilage complex injury in wrist MRI, why not use the existing widely used algorithms, there are too many detection algorithms, as it is classically done in different topics?
  • What are the advantages of the proposed algorithms  in this work over classical and existing algorithms?
  • How this work will be beneficial to doctors for effectively and simply detecting triangular fibrocartilage complex injury in wrist MRI?
  • Simply, why there is a need to your study?

In fact, there are too many detection algorithms in different topics.  A good review is needed for the widely used detection algorithms. There is a significant need to a comparison study between the  existing algorithms and the  proposed algorithms in this paper.

The answers of these logical questions help readers to understand the benefits from this study and how doctors will benefit from its findings.

Reviewer 2 Report

Some notes about the work:
1) The amount of image in the training and test set should be in the text and not just in the image.
2) Only the graphs and values of the ROC curve are present in the results, but the results of the F1 score are missing, for example.
3) Can the results obtained not be compared with the results of related work? Do such works exist?

Overall the work has its merits, but the presentation needs to be substantially improved.

Reviewer 3 Report

Comments to the Authors

  • Page 2 in the MRI datasets section, the 72.8 datasets used in missing percent sign. It was confusing to read without percent.
  • Typo/Grammar error in the Discussion: “The MRNet architecture in this study was a trained 3D CNN with logistic regression to combine the results for the final prediction.”  Remove “a”. Also the meaning is not clear. Do the authors mean “trained on three CNN architectures”?
  • The link to the source code leads "page not found": https://github.com/akousist/tfcc-detection. This raises the question that the code might not be available for a long time if published.
  • The Deep learning architecture used has no novelty. Just implementation of existing deep learning networks. ResNet and AlexNet. This appears more like an implementation manuscript and no novelty is introduced
  • No motivation was given for the choice of given parameters in the model architecture. That is meant to be the focus on validation. But nothing was mentioned in this manuscript.
  • The conclusion of the manuscript is flawed and if this was cited to be the motivation for this work, it is not just strong enough. The authors stated “we confirmed the feasibility of two deep learning algorithms with Res-Net and MRNet backbones to detect hand TFCC injuries using features extracted from MRI scans and compared the performance of the two algorithms. The adopted approach is useful to evaluate MRI features of TFCC injuries”. 

First, many earlier works have shown the ability to use deep learning models to detect features or perform classification from MRI images.  Citations (a) and (b) shows this among many others. I am not sure why the authors think using  TFCC detection from MRI would be different. If there is indeed a justification for this, they should provide it. Hence, if this is the motivation for this work, it is just too plain and uninteresting. This study by  Ng et al, 2017 (citaiton c) did mention that “TFCC tear is difficult to be diagnosed on MRI for its intrinsic small and thin structure with complex anatomy.” Is this why this is worth studying? 

  1. Lu, S., Wang, S. H., & Zhang, Y. D. (2020). Detecting pathological brain via ResNet and randomized neural networks. Heliyon, 6(12), e05625
  2. Kang, J., Ullah, Z., & Gwak, J. (2021). Mri-based brain tumor classification using ensemble of deep features and machine learning classifiers. Sensors, 21(6), 2222.
  3. Ng, A. W., Griffith, J. F., Fung, C. S., Lee, R. K., Tong, C. S., Wong, C. W., ... & Ho, P. C. (2017). MR imaging of the traumatic triangular fibrocartilaginous complex tear. Quantitative imaging in medicine and surgery, 7(4), 443.
  4. Thian, Y. L., Li, Y., Jagmohan, P., Sia, D., Chan, V. E. Y., & Tan, R. T. (2019). Convolutional neural networks for automated fracture detection and localization on wrist radiographs. Radiology: Artificial Intelligence, 1(1), e180001.
  • The authors should compare the performance of their method with other traditional methods or non-deep learning methods for TFCC detection. This way we can see how they perform side by side.
  • The comments listed as future work appear to me to be more current work. For instance, the authors mentioned “ improving the ResNet architecture to improve its detection performance” as a future work. However, questions have not been addressed about why the current architecture did not work. What parameters adjustments did the author make? Where is the plot of the validation?
  • The statistical analysis section is rather presumptuous. There were no plots shown to corroborate the results of the experiments. What were the testing errors and accuracy? On which datasets were the ROC tests perofmred on? As a side note, mentioning the number of engineers and their experience is irrelevant in this section.  
  • Fragments of the content were copied from:
    • https://journals.plos.org/plosmedicine/article?id=10.1371%2Fjournal.pmed.1002699
    • https://www.nature.com/articles/s41380-019-0534-x?code=c38fef59-76d9-4ada-8cd8-f1f6d9ce4dc9&error=cookies_not_supported
    • https://pubs.rsna.org/doi/full/10.1148/radiol.2020190925?
    •